# Clinical Value of Ultrasonography and Serum Markers in Preoperative N Staging of Thyroid Cancer

**DOI:** 10.3390/cells11223621

**Published:** 2022-11-15

**Authors:** Hui Wang, Shanshan Zhao, Chunyang Xu, Jincao Yao, Xiuhua Yu, Dong Xu

**Affiliations:** 1Department of Ultrasound, Joint Service Support Force 903 Hospital, Hangzhou 310013, China; 2Department of Ultrasound, Shaoxing People’s Hospital, Shaoxing 312000, China; 3Department of General Surgery, Joint Service Support Force 903 Hospital, Hangzhou 310013, China; 4Department of Ultrasound, The Cancer Hospital of the University of Chinese Academy of Sciences (Zhejiang Cancer Hospital), Institute of Basic Medicine and Cancer (IBMC), Chinese Academy of Sciences, Key Laboratory of Head & Neck Cancer Translational Research of Zhejiang Province, Hangzhou 310022, China

**Keywords:** lymph node metastasis, neoplasm staging, thyroglobulin, thyroid cancer, ultrasonography

## Abstract

We aimed to determine factors influencing lymph node metastasis (LNM) and develop a more effective method to assess preoperative N staging. Overall, data of 2130 patients who underwent thyroidectomy for thyroid cancer between 2018 and 2021 were retrospectively analysed. Patients were divided into groups according to pN0, pN1a, and pN1b stages. Pathology was used to analyse the correlation between preoperative serum marker indicators and LNM. Receiver operating characteristic curves were used to compare the diagnostic value of ultrasound (US) examination alone, serum thyroglobulin, age, and combined method for LNM. A significant moderate agreement was observed between preoperative US and postoperative pathology for N staging. Between the pN0 and pN1 (pN1a + pN1b) groups, the differences in free triiodothyronine, anti-thyroid peroxidase antibody, and serum thyroglobulin levels were statistically significant. Among the indicators, serum thyroglobulin was an independent predictor of LNM. The area under the receiver operating characteristic curve was 0.610 for serum thyroglobulin level for predicting LNM, 0.689 for US alone, and 0.742 for the combined method. Both preoperative US and serum thyroglobulin level provide a specific value when evaluating the N staging of thyroid cancer, and the combined method is more valuable in the diagnosis of LNM than US alone.

## 1. Introduction

In most countries, the incidence of thyroid cancer (TC) has increased over the last 3 decades [1]. Due to the presence of abundant cervical lymphatic drainage in the area, patients with TC often have central or lateral cervical lymph node metastasis (LNM) [2,3]. LNM is a risk factor for recurrence and survival in patients with TC [2,4,5]. Furthermore, LNM is the basis for N staging and is used to assess the severity of TC. For TC patients with LNM, resection combined with neck lymph node dissection has become a procedure with clinical consensus. However, in patients with uncertain LNM, the significance of prophylactic lymph node dissection as a treatment option remains controversial [6,7].

Ultrasound (US) examination is the best imaging modality for evaluating thyroid nodules [8], which is helpful for the detection of abnormal lymph nodes in the neck. It is the most important imaging method for the preoperative evaluation of N staging of TC and can help clinicians determine whether lymph node dissection is required and also measure the dissection area. Currently, US can detect suspicious metastatic lymph nodes; however, there are still several small metastases that cannot be detected using imaging examinations [9].

A number of studies have shown that preoperative levels of serum thyroid antibodies and those of the thyroid-stimulating hormone (TSH), free triiodothyronine (fT3), free thyroxine (fT4), triiodothyronine (TT3), and thyroxine (TT4) have a certain value in the diagnosis of TC [10,11,12,13,14,15,16]. However, whether they are applicable for the evaluation of N staging of TC is unclear. Only a few studies have shown low serum anti-thyroid peroxidase (TPOAb) concentration to be positively correlated with LNM in patients with TC [17].

In this study, the clinical data of patients with TC were retrospectively analysed to explore the value of preoperative US examination for N staging of TC and to determine whether there is a relationship between serum markers and N staging to provide a basis for clinical diagnoses. This retrospective analysis studied a large variety of serum markers to evaluate the relationship between these serum markers and LNM.

## 2. Materials and Methods

### 2.1. Patients

This multicentre retrospective study included 1983 TC patients from the Cancer Hospital of the University of Chinese Academy of Sciences (Zhejiang Cancer Hospital) and 147 TC patients from Joint Service Support Force 903 Hospital who underwent thyroidectomy between August 2018 and February 2021. We included patients with TC who underwent total or hemi-thyroidectomy and had complete clinical data information. Patients were excluded from the study if they met any of the following criteria: (1) underwent non-curative surgery, (2) underwent reoperation, (3) preoperative serological indicators could not be obtained, or (4) had incomplete clinical data or missing follow-up data. Finally, a total of 1983 patients with 2234 lesions were enrolled in this study. Additionally, the independent predictors in the 147 patients with 160 lesions who met the aforementioned criteria between January 2016 from November 2021 in the Joint Service Support Force 903 Hospital were collected to form the external validation group. The Medical Ethics Committee of Zhejiang Cancer Hospital (approval number: IRB-2020-287) and Joint Service Support Force 903 Hospital (approval number: 20211224/11/01/001) approved informed consent waiver.

### 2.2. Ultrasonic Instruments and Inspection Methods

GE Logiq E9 [General Electric (GE) Healthcare, Chicago, IL, USA], Toshiba APLIO 400 TUS-A400 (Toshiba Medical Systems, Otawara, Japan), Hitachi Arietta70 (Hitachi, Tokyo, Japan), and other colour Doppler US detectors were used. The probe frequency was 5–15 MHz. All patients underwent pre-surgical colour Doppler US examinations performed by a diagnostician with more than 3 years of experience in thyroid imaging. If there was any doubt, the deputy chief physician was requested to make a final decision. Patients were placed in the supine position, with the neck fully exposed. The appearance of the lesion was evaluated carefully using US, including the size, location, shape, calcification, and boundary. If the nodules appeared solid, very hypoechoic, microcalcification, and taller-than-wide, they would likely be malignant [18]. Then, the cervical lymph nodes were also assessed carefully. Sonographic features suggestive of abnormal metastatic lymph nodes included enlargement, loss of the fatty hilum, a rounded rather than an oval shape, hyperechogenicity, cystic change, calcification, and peripheral vascularity [18]. This evaluation was based on the 2015 American Thyroid Association (ATA) management guidelines for adult patients with thyroid nodules.

### 2.3. Type of Surgery

All patients underwent prophylactic central compartment lymph node dissection performed by three experienced head and neck surgeons. For the lateral cervical lymph nodes, if the US results showed suspicious metastatic lymph nodes, the lymph nodes in the region were cleared. A total of 362 patients underwent therapeutic lateral cervical lymph node dissection (18.3%). All lymph nodes underwent postoperative pathological examination.

### 2.4. TNM Staging Criteria for TC

According to the eighth edition of the American Joint Committee on Cancer (AJCC) TC staging system [19], N staging is defined as follows: N0, no evidence of locoregional LNM; N1, metastasis to regional nodes; N1 includes the N1a and N1b lymph nodes (N1a, metastases to level VI or VII [pre-tracheal, paratracheal, pre-laryngeal/Delphian, or upper mediastinal]; N1b, metastasis to unilateral, bilateral, or contralateral lateral neck lymph nodes [levels I, II, III, IV, or V], or retropharyngeal lymph nodes). In this study, N staging detected with US examination was defined as clinical N staging.

### 2.5. Detection of Serum Markers

The TT3 (reference range: 0.66–1.92 ng/mL), TT4 (reference range: 4.3–12.5 μg/dL), fT3 (reference range: 1.80–4.10 pg/mL), fT4 (reference range: 0.81–1.89 ng/dL), TSH (reference range: 0.380–4.340 μIU/mL), anti-thyroglobulin (TgAb, reference range: 0.0–60.0  IU/mL), and TPOAb (reference range: 0.0–60.0 IU/mL) levels were measured using an ADVIA Centaur automatic chemiluminescence analyser (Siemens, Munich, Germany) 1–3 days before surgery. The fasting serum thyroglobulin (Tg, reference range: 1.40–78 ng/mL) level was measured using Cobase 8000 E602 (Roche Diagnostics, Basel, Switzerland) at the same time. If the test value exceeded the detection range of the instrument, the group spacing was estimated using the interval grouping method, and the value was replaced.

### 2.6. Statistical Analyses

SPSS version 25.0 statistical software (IBM Corporation, Armonk, NY, USA) was used to analyse the data. The count data are expressed as cases, and agreement between the consistency of US examination findings and pathological findings was determined using the kappa statistic. Kappa values between 0.21 and 0.40 indicated general consistency; 0.41 and 0.60, medium consistency; and 0.61 and 0.80, high consistency. The measurement data were tested using the Shapiro–Wilk test and Kolmogorov–Smirnov test for normality. The measurement data (non-normal distribution) are expressed as medians and quartiles. The comparison between the groups was analysed using the Wilcoxon rank-sum test. Factors with *p* < 0.05 were selected, and multivariable logistic regression analysis was performed. Using forward stepwise selection, the independent associated factors were screened out according to the likelihood ratio test (*p* < 0.05). At the same time, the regression coefficient, odds ratios (ORs), and 95% confidence intervals (CIs) were calculated. ORs were used to estimate the regression coefficients of these independent factors to calculate the combined method. The receiver operating characteristic (ROC) curves of independent factors and the combined method for LNM were constructed. Youden’s index was used to determine the optimal cut-off value and its corresponding sensitivity and specificity. Simultaneously, the area under the ROC curve (AUC) was calculated. The combined method was used in the external verification groups to verify its accuracy.

## 3. Results

### 3.1. Patient Characteristics

A total of 1983 patients, including 539 male and 1444 female patients, with TC confirmed by postoperative pathology were enrolled retrospectively. The mean age of the patients was 45.26 ± 12.34 years. The external validation group consisted of 147 patients, including 41 males and 106 females, with a mean age of 45.12 ± 12.08 years. We further divided the patients into N0, N1a, and N1b groups, according to the eighth edition of the AJCC TC staging system [19] and pathology reports. Table 1 summarises the demographic and clinical characteristics of the patients with TC (Table 1).

### 3.2. Pathological and Ultrasonic Diagnostic Results of LNM

Among 1983 patients, the pathological results of 873 patients were stage N1 (pN1), including 571 (28.8%) cases of stage pN1a and 302 (15.3%) cases of stage pN1b. The pathological results of 1110 (56.0%) patients were stage N0 (pN0). The US examination results identified 1360 cases of stage N0 (cN0), 261 of stage cN1a, and 362 of stage cN1b. There was a medium consistency between the US examination for N staging and pathological staging (*κ* = 0.437, *p* < 0.01) (Table 2).

### 3.3. Comparison of Serum Markers between the pN0 and pN1 Groups

According to the Shapiro–Wilk and Kolmogorov–Smirnov normality tests, none of the serum marker indexes conformed to a normal distribution; thus, the Wilcoxon rank-sum test was used. Between the pN0 and pN1 groups, the differences in fT3, TPOAb, and Tg levels were statistically significant (all *p* < 0.05). There was no significant difference in the TT3, TT4, fT4, TSH, and TgAb levels between the pN0 and pN1 groups (all *p* > 0.05). The median values of serum markers in both groups are shown in Table 3.

### 3.4. Analysis of the Correlation of fT3, Tg, and TPOAb with the pN0 and pN1 Groups

A multivariable logistic regression was used to analyse the association of the fT3, TPOAb, and Tg levels with LNM. The results showed that in the US diagnosis of LNM, Tg, TPOAb, and age were independent predictors of LNM (all *p* < 0.05). The effect of the fT3 level on LNM was not statistically significant (*p* = 0.256) (Table 4). However, the 95% CI for OR of TPOAb was 0.999–1.000, indicating that TPOAb was not an independent predictor of LNM.

Age (OR = 0.973, *p* < 0.001), US diagnosis of LNM (OR = 6.357, *p < 0.001*), and Tg level (OR = 1.003, *p* < 0.001) were independent risk predictors of LNM in TC. We established a predictive model based on the following multivariable logistic regression: Logit (*p*) = −0.058 + 0.003 × Tg—0.027 × age + 1.849 × cN.

### 3.5. Comparison of the Diagnostic Value of Tg, Age, US Examination Alone, and the Combined Method in LNM

We performed a ROC curve analysis of three risk predictors (Tg levels, age, and US examination alone) and the combined method in predicting LNM (Figure 1). The AUCs of Tg levels, age, US examination alone, and the combined method were 0.610 (95% CI, 0.584–0.635), 0.614 (95% CI, 0.589–0.638), 0.689 (95% CI, 0.665–0.713), and 0.742 (95% CI, 0.720–0.764), respectively (Figure 1). The best cut-off point for Tg was 17.78 ng/mL, and the best cut-off point for age was 40.5 years. The AUC of the combined method was 0.742, demonstrating that the combined method had a satisfactory classification effect on LNM. The specificity and sensitivity were 80.0% and 59.5%, respectively. The AUC of the combined method was higher than that of the US examination alone.

### 3.6. Verification of the Diagnostic Value of Tg, Age, US Examination Alone, and Combined Method in LNM

The AUCs of Tg levels, age, US examination alone, and the combined method in the external validation group were 0.599 (95% CI, 0.505–0.693), 0.633 (95% CI, 0.537–0.728), 0.621 (95% CI, 0.526–0.716), and 0.700 (95% CI, 0.612–0.789), respectively (Figure 2).

### 3.7. Comparison of TPOAb and Tg between the pN1a and pN1b Groups

The Wilcoxon rank-sum test was performed to obtain Z = −7.596 (*p* < 0.001), indicating that there were significant differences in the Tg level between the pN1a and pN1b groups, which was also significant for the preoperative prediction of LNM in the central region or the lateral neck. However, the differences in the TPOAb levels between the pN1a and pN1b groups were not statistically significant (*p =* 0.404).

## 4. Discussion

To the best of our knowledge, this study is the first to use the largest variety of serum markers associated with LNM in TC. A total of eight serum markers in 2130 patients with TC were analysed. Using the Wilcoxon rank-sum test and multivariate analysis, we found that the US diagnosis of LNM, Tg levels, and age were independent predictors of LNM in patients with TC. Their combination method has the best prediction effect. The external validation group also confirmed the effect of Tg and age in predicting LNM.

Recently, the rapidly increasing incidence of TC has become a global problem [20]. Differentiated TC (DTC), which includes papillary TC (PTC) and follicular cancer, comprises the vast majority (>90%) of all TCs [21]. Therefore, most of the studies on TC have focused on DTC and PTC. This study did not distinguish between the different pathological types because the purpose was to predict N staging with preoperative data, and in most cases, the pathological type of TC cannot be determined before surgery.

Since DTC accounts for the majority of TC cases and has a good prognosis and high treatment rate, clinicians do not pay enough attention to preoperative staging, resulting in poor surgical effects in some patients and even the need for repeat operation in others. Therefore, accurate preoperative staging evaluation can play an instructive role in guiding the selection of treatment options and the evaluation of the prognosis in patients. This will help to avoid the uncertainty of surgical methods caused by inaccurate preoperative staging and, thus, enhance the effect of surgical treatment, improve the clinical prognosis, and reduce the rate of reoperation. Determining whether LNM is present, and the degree of metastasis, is of great significance as independent indicators of the N stage. On one hand, tumour metastases in TC to the lymph nodes increase the risk of recurrence of disease and are associated with poor prognosis [22]. Thus, metastatic lymph nodes should be treated actively. On the other hand, an increasing number of thyroid nodules have been found in unrelated imaging studies, leading to an increased diagnosis of low-risk TC. Therefore, there is a greater emphasis on risk assessment based on clinical and sonographic features to avoid morbidity secondary to unnecessary therapy [23]. Guo et al. [24] indicated that whether lymph node dissection should be performed or not usually depends on the preoperative determination of LNM. Therefore, determining the presence of LNM as early as possible is of great significance for improving patient prognosis.

US examinations may provide thorough evaluation of nodules, capsular invasions, and LNM, which can significantly influence the extent of surgical resections [25,26]. In most cases, for lateral cervical lymph nodes, only when LNM was suspected or found on preoperative US or computed tomography examinations, the surgeon would have treated the lymph nodes. US examinations regard enlargement, loss of the fatty hilum, a rounded rather than an oval shape, hyperechogenicity, cystic change, calcification, and peripheral vascularity as the characteristics of suspicious lateral cervical metastatic lymph nodes, and the results showed a high coincidence rate with pathological results. Lateral neck lymph node dissection is associated with trauma and adverse effects, and the accuracy of US diagnosis of lateral cervical lymph nodes is greater than that of central lymph nodes, which then eliminates the necessity of prophylactic dissection [27]. This study showed a medium consistency between US for N staging and the actual pathological staging. On one hand, US examination is susceptible to bone, gas, and other interferences, and it is easy to miss some lymph nodes in special parts, such as the paratracheal and supraclavicular regions. On the other hand, it is sometimes difficult to distinguish metastatic lymph nodes from inflammatory lymph nodes in TC using US images, especially in the central lymph nodes. Leboulleux et al. [28] indicated that an accurate diagnosis of LNM was challenging in clinical practice, with only approximately 40% being detected before surgery. In another study in 2007, Leboulleux et al. [29] indicated that preoperative US examinations identified only half of the lymph nodes found during surgery due to the presence of the overlying thyroid gland. Therefore, central lymph nodes are more difficult to detect than lateral cervical lymph nodes.

Zhao et al. [27] indicated that the preoperative US examination had poor sensitivity in the diagnosis of central LNM, while good diagnostic efficacy was shown for lateral LNM of PTC. Prophylactic central lymph node dissection is recommended for patients with PTC due to the high incidence of central LNM and low diagnostic efficacy of US examinations.

Thus, a preoperative assessment of LNM using US examinations alone cannot be used as the only basis for the surgical treatment of lymph nodes; however, it is controversial whether routine prophylactic central lymph node dissections should be performed for TC [6,7,30,31]. Based on the mechanisms of tumorigenesis and progression, the treatments for cancer have become increasingly sophisticated and standardised [32]. In some studies [33,34,35,36,37], preventive cleaning has shown no improvement in long-term outcomes, while increasing the likelihood of morbidity and side effects such as hypocalcaemia, recurrent nerve injury, and hypoparathyroidism. Therefore, more accurate methods are needed to predict LNM in patients with TC. In this study, we collected data on LNM from US examinations, serum markers, and age of patients with TC to obtain a more comprehensive analysis using these new factors. At the same time, the combined method was used to help clinicians predict LNM before surgery, which may be useful to determine the scope of surgery and further improve quality of life in patients.

The 2015 ATA guidelines [18] suggested that serum TSH levels should be measured when thyroid nodules greater than 1 cm in diameter are found. A number of studies have shown that preoperative serum thyroid antibody levels and TSH, fT3, fT4, TT3, and TT4 levels have value in assessing benign and malignant thyroid nodules [11,12,13,14,15,16]. Some studies on Tg point out that the routine measurement of Tg levels or the initial evaluation of thyroid nodules is not recommended. Serum Tg levels can be elevated in most thyroid diseases, and the test is insensitive and nonspecific for TC [38]. However, other studies have suggested that the preoperative serum Tg level may play a role in differentiating between benign and malignant thyroid nodules [39,40]. The 2015 ATA guidelines [18] also suggested that the postoperative serum Tg level (on thyroid hormone therapy or after TSH stimulation) can help in assessing the persistence of the disease or thyroid remnants and predicting potential future disease recurrence. Several studies [41,42,43] have indicated the value of the serum Tg level in the postoperative management of patients with TC. However, there are few studies on the use of these serum markers to determine whether TC involves LNM. When thyroid US examination suggests malignant thyroid nodules, but the determination of the nature of the lymph nodes is difficult, can the benign and malignant lymph nodes be differentiated with the use of serum markers? The formation and development of the tumour involve the comprehensive action of internal and external factors [44]. Based on the findings of these previous reports, our study collected data on the serum markers from preoperative routine examinations and evaluated the differences between the pN0 and pN1 groups. In patients without LNM and with LNM, the TT3, TT4, fT4, TSH, and TgAb levels showed no statistically significant differences. Although these indicators were not helpful in determining whether LNM occurred in patients, the other three markers, fT3, TPOAb, and Tg, showed significant differences. Next, we performed a multivariate logistic regression analysis. The results showed that the US diagnosis of LNM, Tg levels, and age were independent predictors of LNM. In the comparison between the pN1a and pN1b groups, only the Tg levels were significantly different; thus, among these serum markers, only the Tg level was applicable for the N staging. The results are not in conflict with the guidelines; the preoperative Tg level evaluation for the assessment of LNM to determine TC severity has not yet been incorporated into the guidelines.

The ROC curves constructed in this study suggested that Tg levels, age, and US examination alone were significant predictors for the diagnosis of LNM of TC (*p* < 0.05); although, the combined method was the best. In clinical practice, age and Tg level can be combined for lymph node enlargement, which cannot be determined using US examination. If the Tg level is elevated abnormally and the patient is younger, LNM may be considered, and central lymph node dissection recommended. If the Tg level is below the critical value, a benign possibility may be considered. Figure 3 and Figure 4 show two cases of suspicious metastatic lymph nodes indicated by US examinations. The Tg level was below the critical value, and the final pathology indicated lymphadenitis.

Tg is a dimeric glycoprotein released by normal follicular tissue and DTC [45]. The role of the preoperative Tg level in the diagnosis and prediction of recurrence and metastasis of TC has not been clarified in the 2015 ATA guidelines [18]. However, in recent years, some scholars have affirmed the value of using preoperative Tg levels. They believe that serum Tg levels can be used not only to monitor disease progression in patients with TC during follow-ups but also as auxiliary markers for diagnosis and risk classification, which has great potential research value [22,46,47,48,49]. This study provided evidence for this option. In addition, Li et al. [17] proposed that TgAb level was not a significant risk factor for LNM. Hu et al. [50] proposed that there was no significant difference in the expression level of TSH, TPOAb, T3, T4, FT3, and FT4 between the pN0 and pN1 groups. The conclusions of this study are consistent with their findings. This study suggested that Tg levels were correlated with LNM, and that there were also significant differences between the pN1a and pN1b groups, indicating that the Tg level was also significant in distinguishing LNM in the central region and LNM in the lateral neck. This is consistent with the research results of Rui et al. [51]. This study suggested that the reason for the promotion of LNM by elevated Tg might be that thyroglobulin was only produced by thyroid follicular cells and involved in thyroid hormone synthesis and iodine transport. Elevated serum Tg usually occurs when thyroid tissue is stimulated or damaged by inflammation or increased tumour load [52]. Both LNM and Tg represent tumour progression, hence elevated Tg is the influencing factor of LNM.

In the Tg assays, the interference with Tg measurement by the presence of anti-Tg and heterophile antibodies, as well as the use of different detection methods, may give rise to varying results in terms of sensitivity or detection limits [18,53]. Currently, in China, the use of radio-immunoassays and immunometric assays have been discontinued, and most laboratories use immunochemiluminometric assays. In this study, TgAb is believed to reduce Tg; thus, high serum Tg levels in the presence of TgAb are more likely to indicate LNM. However, the interference of the antibodies cannot be avoided. Therefore, in clinical work, we need to exclude some Tg false positive cases. When thyroglobulin elevation does not match up with the clinical picture, then we should look at the performance of the assay and consider the presence of heterophile antibodies. It also highlights the essential nature of close communication between surgeons, clinicians and biochemists in order to avoid unnecessary diagnostic procedures and treatments.

Multiple studies have shown that younger TC patients, have a higher risk of LNM [54,55,56,57]. This study suggests that TC patients younger than 41 years have a higher risk of LNM and should be treated more aggressively. However, this should not include children younger than 18, and we have too few data on pediatric patients to draw accurate conclusions. Spinelli et al. [58] indicated that thyroidectomy can be associated with a considerable rate of complications such as hypoparathyroidism and recurrent laryngeal nerve injury. For all of these reasons, they underline the necessity to develop specific management strategies for the pediatric population, in order to face the increasing rates of thyroid cancers in the pediatric field and to offer optimal surgical treatment and avoid unnecessary invasive approaches. For pediatric patients, since US has higher sensitivity for lateral neck node metastasis than the central one, we do not recommend a prophylactic lymph node dissection in patients clinically negative for LNM. A therapeutic lymph node dissection should be performed only in patients with pre-operative evidence of central and/ or lateral neck metastasis [59,60].

The limitations of this study are as follows. (1) The results of the study may not be applicable to other races and countries because the patients that were included in this study were all from China. (2) Due to the large sample size, different brands of multiple ultrasonic instruments were used; therefore, the image presentations may be inconsistent. (3) The US analysis may have been affected by subjective factors, and the description of the lymph nodes using different phrases may have led to different results when using US for N staging. (4) Tg is also influenced by other interference factors, such as the volume of the gland, TSH values, and iodine content of the residential area. Although these interference factors are likely to have influence on the correctness of Tg critical value, they have not been accounted for in this study.

## 5. Conclusions

It is necessary to identify the metastatic lymph nodes not detected by US examinations to prevent misdiagnoses. If suspicious lymph nodes found by US examinations can be determined before surgery, the excessive dissection of normal lymph nodes may be avoided. Specifically, when the preoperative Tg level is abnormally high and the patient is 18–41 years old, it is necessary to perform a central lymph node dissection to avoid missing lymph nodes not detected with US examinations. When the preoperative Tg level is lower than the critical value and the patient is older than 41 years, a comprehensive evaluation is needed for the suspected lymph nodes detected using US examinations, which may help to avoid excessive dissection of normal lymph nodes. This study found that the diagnostic sensitivity and specificity of the combined method for the detection of LNM was higher than those of US examination alone. In view of the extent of the correlation, it is suggested that preoperatively, clinicians use US examination, age, and Tg level to assess LNM to make relatively accurate predictions, to determine whether lymph node dissection is needed, as well as the scope of dissection.

## Figures and Tables

**Figure 1 cells-11-03621-f001:**
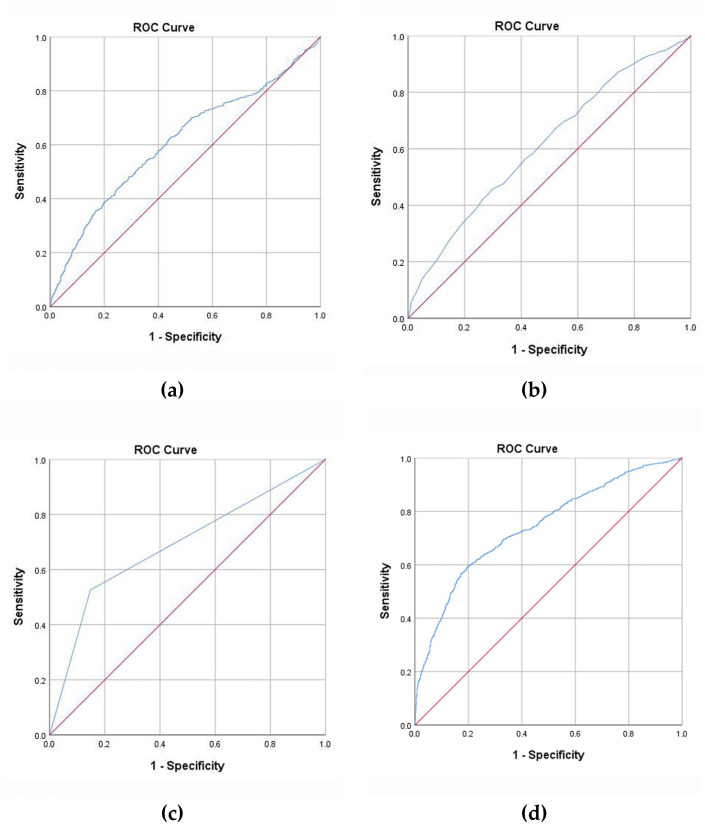
Risk factors and the combined method for predicting LNM in the model establishing group. (**a**) Tg. (**b**) Age. (**c**) US alone. (**d**) Combined diagnosis. LNM, lymph node metastasis; ROC, receiver operating characteristic; Tg, serum thyroglobulin; US, ultrasound.

**Figure 2 cells-11-03621-f002:**
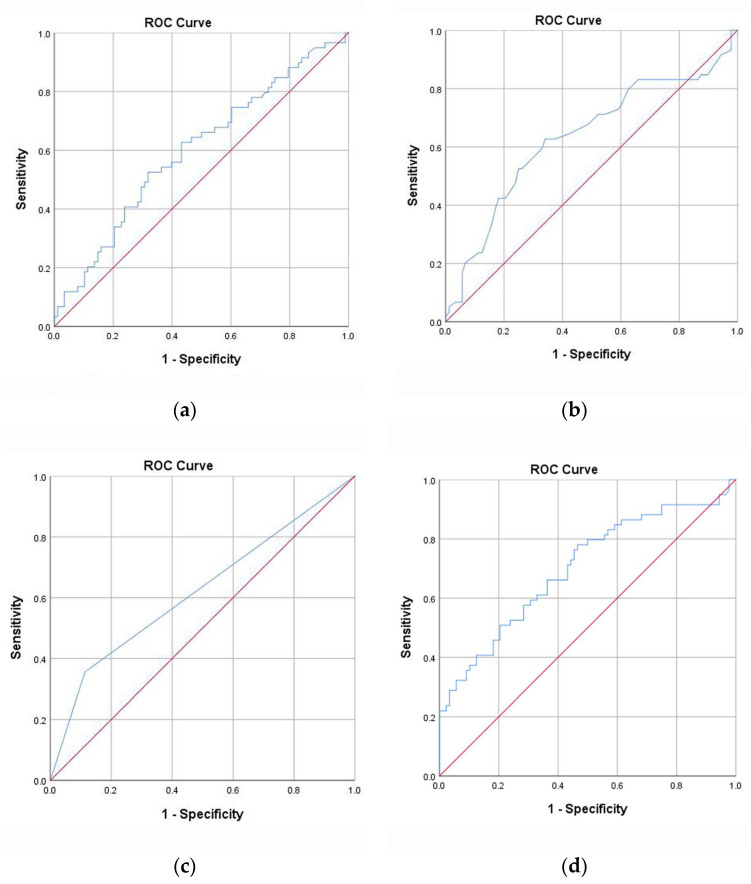
Risk factors and the combined method for predicting LNM in the external validation group. (**a**) Tg. (**b**) Age. (**c**) US alone. (**d**) Combined diagnosis. LNM, lymph node metastasis; ROC, receiver operating characteristic; Tg, serum thyroglobulin; US, ultrasound.

**Figure 3 cells-11-03621-f003:**
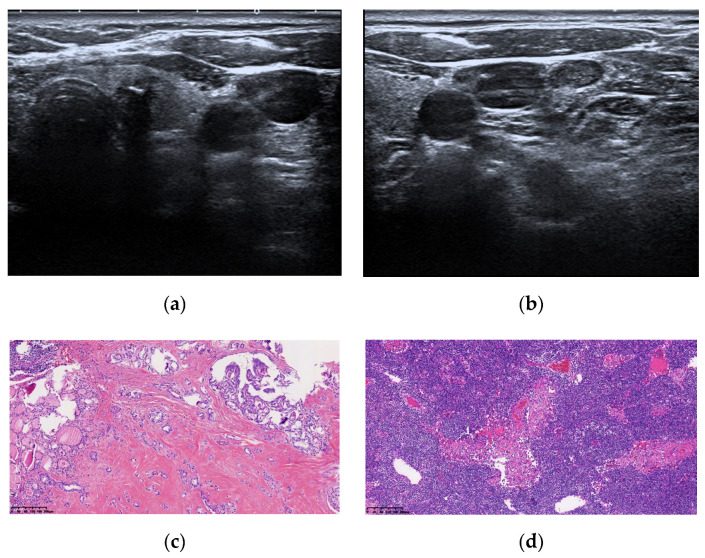
A 57-year-old female patient with a preoperative serum Tg level of 2.95 ng/mL. Ultrasound findings: thyroid left lower lobe nodule, TI-RADS 5 categories (**a**) Left neck IV lymph node enlargement, suspicious metastatic lymph nodes (**b**) combined with puncture if necessary. Postoperative pathology shows papillary thyroid microcarcinoma ((**c**) shows HE staining, magnification ×100). Left cervical lymph node chronic inflammation (**d**) shows HE staining, magnification ×100). Tg, serum thyroglobulin; HE, haematoxylin and eosin; TI-RADS, Thyroid Imaging Reporting and Data System.

**Figure 4 cells-11-03621-f004:**
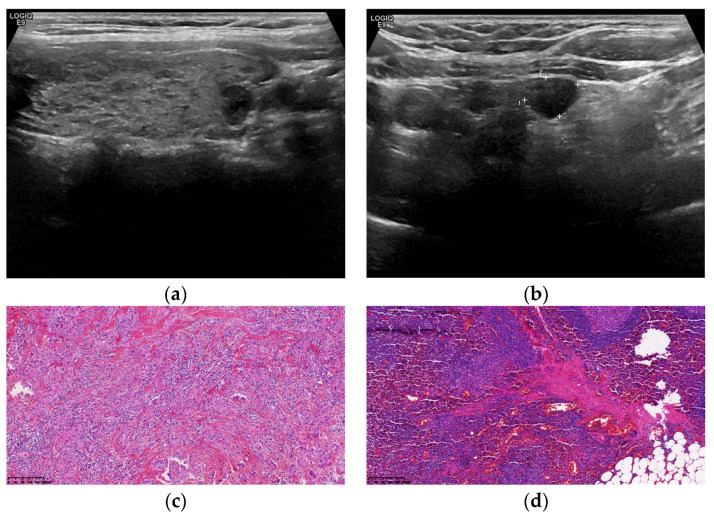
A 35-year-old female patient with a preoperative serum Tg of 2.31 ng/mL. Ultrasound findings: thyroid left lower lobe nodule, TI-RADS 5 categories (**a**) Left cervical lymph node enlargement, suspicious metastatic lymph nodes (**b**) Postoperative pathology shows papillary thyroid microcarcinoma (**c**) shows haematoxylin and eosin [HE] staining, magnification ×100). Left cervical lymph node chronic inflammation (**d**) shows HE staining, magnification ×100). Tg, serum thyroglobulin; HE, haematoxylin and eosin; TI-RADS, Thyroid Imaging Reporting and Data System.

**Table 1 cells-11-03621-t001:** Demographic and clinical characteristics of patients in the model establishing and external validation groups.

Characteristics	Model Establishing Group			External Validation Group		
	N0 (%)	N1a (%)	N1b (%)	N0 (%)	N1a (%)	N1b (%)
Sex						
Male	261 (23.5)	173 (30.3)	105 (34.8)	17 (19.3)	18 (38.3)	6 (50.0)
Female	849 (76.5)	398 (69.7)	197 (65.2)	71 (80.7)	29 (61.7)	6 (50.0)
Age, median (IQR), years	48 (19)	43 (18)	41 (21)	48 (16)	42 (18)	34 (30)
Pathology ^1^						
Papillary	1075 (96.8)	566 (99.1)	293 (97.0)	85 (96.6)	47 (100)	12 (100)
Follicular	18 (1.6)	1 (0.2)	1 (0.3)	1 (1.1)	0 (0)	0 (0)
Medullary	15 (1.4)	4 (0.7)	8 (2.6)	2 (2.3)	0 (0)	1 (8.3)
Undifferentiated	2 (0.2)	0 (0)	1 (0.3)	0 (0)	0 (0)	0 (0)
Serum markers						
TT3, median (IQR), ng/mL	1.07 (0.23)	1.07 (0.23)	1.09 (0.22)	0.995 (0.21)	1.03 (0.27)	0.98 (0.22)
TT4, median (IQR), μg/dL	7.80 (2.00)	7.70 (1.80)	7.90 (2.00)	8.10 (2.12)	8.40 (2.34)	7.85 (1.95)
fT3, median (IQR), pg/mL	3.20 (0.42)	3.23 (0.48)	3.21 (0.47)	3.15 (0.50)	3.24 (0.66)	3.275 (0.46)
fT4, median (IQR), ng/dL	1.24 (0.22)	1.24 (0.24)	1.27 (0.24)	1.095 (0.22)	1.15 (0.28)	1.165 (0.19)
TSH, median (IQR), μIU/mL	1.55 (1.18)	1.54 (1.07)	1.78 (1.36)	1.46965 (0.7965)	1.368 (1.225)	1.522 (0.6345)
TgAb, median (IQR), IU/mL	15.60 (15.62)	15.90 (15.65)	15.00 (18.35)	14.00 (34.82)	14.00 (46.28)	17.60 (133.03)
TPOAb, median (IQR), IU/mL	38.55 (29.80)	33.70 (22.70)	31.30 (24.65)	26.20 (48.24)	26.20 (40.09)	26.20 (3.63)
Tg, median (IQR), ng/mL	10.43 (14.43)	13.89 (19.57)	31.55 (72.26)	8.105 (13.59)	10.24 (18.80)	28.09 (79.67)

IQR, interquartile range; TT3, triiodothyronine; TT4, thyroxine; fT3, free triiodothyronine; fT4, free thyroxine; TSH, thyroid-stimulating hormone; TgAb, anti-thyroglobulin; TPOAb, anti-thyroid peroxidase; Tg, thyroglobulin. ^1^ Patients who have two pathological types.

**Table 2 cells-11-03621-t002:** Preoperative ultrasound N staging and pathological N staging results of patients with thyroid cancer.

Clinical N Stage	Pathological N Stage			Sum	Kappa	*p*-Value
	pN0	pN1a	pN1b			
cN0	946	405	9	1360	0.437	<0.001
cN1a	125	133	3	261		
cN1b	39	33	290	362		
Sum	1110	571	302	1983		

**Table 3 cells-11-03621-t003:** Comparison of serum marker results between the pN0 and pN1 groups.

Serum Markers	Median		Z	*p*-Value
	pN0	pN1		
TT3 (ng/mL)	1.070	1.070	−0.342	0.733
TT4 (μg/dL)	7.800	7.700	−1.249	0.212
fT3 (pg/mL)	3.200	3.220	−2.789	0.005
fT4 (ng/dL)	1.240	1.250	−1.212	0.226
TSH (μIU/mL)	1.547	1.611	−1.039	0.299
TgAb (IU/mL)	15.600	15.600	−0.454	0.650
TPOAb (IU/mL)	38.550	32.800	−5.212	<0.001
Tg (ng/mL)	10.430	16.580	−8.399	<0.001

TT3, triiodothyronine; TT4, thyroxine; fT3, free triiodothyronine; fT4, free thyroxine; TSH, thyroid-stimulating hormone; TgAb, anti-thyroglobulin; TPOAb, anti-thyroid peroxidase; Tg, thyroglobulin.

**Table 4 cells-11-03621-t004:** Multivariable logistic regression analysis results of Tg, TPOAb, fT3 levels, age, and cN.

Variables	β	SE	Wald	*p*-Value	OR	95% CI for OR
Constant	−0.058	0.493	0.014	0.906	0.944	
TPOAb (IU/mL)	−0.001	0.000	25.651	<0.001	0.999	0.999–1.000
Tg (ng/mL)	0.003	0.001	17.276	<0.001	1.003	1.001–1.004
fT3 (pg/mL)	0.147	0.130	1.288	0.256	1.158	0.899–1.494
Age	−0.027	0.004	39.817	<0.001	0.973	0.965–0.981
cN (0 or 1)	1.849	0.115	257.547	<0.001	6.357	5.071–7.967

SE, standard error; OR, odds ratio; CI, confidence interval; TPOAb, anti-thyroid peroxidase; Tg, thyroglobulin; fT3, free triiodothyronine; cN, clinical N stage.

## Data Availability

The data presented in this study are available on request from the corresponding author. The data are not publicly available due to privacy.

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
