# Peer review of "Clinical Value of Ultrasonography and Serum Markers in Preoperative N Staging of Thyroid Cancer"

_cells, 2022, doi:10.3390/cells11223621_

Round 1
Reviewer 1 Report
The present paper has an important impact for surgical strategies.
Although some minimal editing should be done.
The paper refer to young patients with a critical value considered as 41 years. We have to keep in mind that differentiated thyroid carcinoma has a different behaviour in pediatric age hence a different therapeutic strategies, so it would be interesting to define a better stratification of the patients according to age considering pediatric age as whole (below and above 18 years of age).
Moreover it would be important to critically review the paper in the light of pediatric differentiated thyroid carcinoma consensus edited by Howard SR et al 2022.
As you mentioned in the introduction section the steady increase of thyroid carcinoma over the past decades and LMN as a risk factor for surgical outcome, I suggest the authors to have a deep look into
"Increased trend of thyroid cancer in childhood over the last 30 years in EU countries: a call for the pediatric surgeon", European Journal of Pediatrics 2022.
and
"Cervical Lymph Node Metastases of Papillary Thyroid Carcinoma, in the Central and Lateral Compartments, in Children and Adolescents: Predictive Factors", World Journal of Surgery, 2018.
The manuscript is well written and it is worth for publication following minor alterations.
Reviewer 2 Report
Clinical Value of Ultrasonography and Serum Markers in Pre-operative N Staging of Thyroid Cancer
Hui Wang , Shanshan Zhao , Chunyang Xu , Jincao Yao , Xiuhua Yu and Dong Xu
I enjoyed reading this paper which attempts to relate a number of different serological markers to affect on-going assessment of post ablation thyroid cancer patients. I think the authors have a produced an important and comprehensive paper that demonstrate some very useful pointers to an increasing problem worldwide.
As I read through it, the limitations that occurred to me were exactly the same ones that the authors highlight at the end of their work. If this can be repeated with different populations of both patients and ultrasonographers, that will increase the usefulness of this work, although personally, I suspect the messages will apply equally to all people.
I would make a couple of observations though and these are not criticisms. I’ve seen reports that anti-TG antibodies can cause a falsely raised TG estimate i.e., it is not just a false negative problem. My point is that, in general terms, when thyroglobulin elevation doesn’t match up with the clinical picture, then we should look at the performance of the assay and consider the presence of heterophile antibodies. It also highlights the essential nature of close communication between surgeons, clinicians and biochemists in order to avoid unnecessary diagnostic procedures and treatments. I would have said that if the TG is absent and anti-TG persists for longer than 6 months post ablation, a new neck ultrasound would be useful.
The paper is comprehensive, the conclusions valid and important, the references are up to date and the English editing is better than mine!! I don’t often review papers in which I would say nothing needs to be changed but this is an exception – Nicely done, folks!
